# Learning the Connections in Direct Feedback Alignment

## Abstract

Feedback alignment was proposed to address the biological implausibility of the backpropagation algorithm which requires the transportation of the weight transpose during the backwards pass. The idea was later built upon with the proposal of direct feedback alignment (DFA), which propagates the error directly from the output layer to each hidden layer in the backward path using a fixed random weight matrix. This contribution was significant because it allowed for the parallelization of the backwards pass by the use of these feedback connections. However, just as feedback alignment, DFA does not perform well in deep convolutional networks. We propose to learn the backward weight matrices in DFA, adopting the methodology of *Kolen-Pollack learning*, to improve training and inference accuracy in deep convolutional neural networks by updating the direct feedback connections such that they come to estimate the forward path. The proposed method improves the accuracy of learning by direct feedback connections and reduces the gap between parallel training to serial training by means of backpropagation.

## 1 Introduction

When feedback alignment was proposed by Lillicrap et al. (2016) it was cited as being a biologically plausible alternative to the backpropagation algorithm, but not long after Nøkland (2016) showed that variants of this approach may show tangible benefits during training such as mitigating the vanishing gradients issue or enabling parallelization of the backwards pass at the cost of additional memory requirements. Recently, interest in the latter has begun to grow as the memory capacity and compute capability of modern GPUs has continued to observe significant leaps. While many of these recently proposed alternatives have been shown to be just as capable as the backpropagation algorithm in terms of inference accuracy on deep convolutional networks, it should be noted that many of these approaches have not yet been shown to perform well outside of the image classification task. Direct feedback alignment (DFA), an earlier approach proposed by Nøkland (2016), was shown to perform reasonably well on a number of natural language processing tasks with recurrent neural networks and transformers by Launay et al. (2020). However, direct feedback alignment still shows poor performance on the image classification task due to its inability to effectively train convolutional layers.

We propose a modification to the DFA algorithm to improve its ability in training deep convolutional neural networks. Due to its relationship with another approach(Akrout et al., 2019), we call our method *Direct Kolen-Pollack* learning or DKP. We empirically show the mechanisms that allow the improvement in our approach over DFA by measuring DKP's ability to better estimate the backpropagation algorithm. We also show this improvement directly by training two deep convolutional neural network architectures on the Fashion-MNIST, CIFAR10, CIFAR100, and TinyImageNet200(Le & Yang, 2015) datasets. More so, we recommend training procedures for training with DFA, pointing out the important role batch normalization plays in our experiments. And while a couple of works have shown that direct feedback connections can be viable when connecting to only the output of a block of layers in a network (Ororbia et al., 2020; Han & Yoo, 2019), we show advances in the case of having feedback connections to *all* layers in deep convolutional neural networks. While direct feedback connections to all layers for current PC hardware, and also from a software perspective, may not be practical, in the future it may be useful for edge devices, IoT, SOC design, *etc*.(Frenkel et al., 2019; Han & Yoo, 2019), especially those that involve learning vision tasks. Thus, making advances in the training scenario of direct feedback connections to all layers

in a neural network at a minimal computational cost for vision tasks is a primary motivation of this work.

## 1.1 RELATED WORK

Alternatives to the backpropagation algorithm have been proposed for their heightened biologically plausibility, or often as a means of parallelizing the training process. More recently, a number of algorithms have shown impressive results on large classification datasets such as ImageNet(Akrout et al., 2019; Kunin et al., 2020; Belilovsky et al., 2019; Xu et al., 2020). To enable further parallelization of the training process, these works often focus on tackling three major deficiencies with the backpropagtion algorithm: *forward locking*, *backward locking*, and *update locking*. Forward locking prevents any calculation of gradients until the forward pass has been completed. Backward locking means that the gradients at some layer can not be calculated until the learning signals at all of the downstream layers have been calculated first. Update locking means that the parameters at some layer cannot be updated before the learning signal at the layer upstream of it has been calculated.

Difference Target Propagation (DTP), proposed by Lee et al. (2015), is one such alternative to the backpropagation algorithm that instead of computing gradients at each layer computes targets that are propagated backwards through the network by means of layer-wise autoencoders. In a recent paper by Lillicrap et al. (2020), DTP and methods that use layer-wise autoencoders in the backward path to propagate gradients are claimed to be more biologically plausible alternative to backpropagation and help to explain how biological neural networks might learn using a process similar to the backpropagation algorithm. Around the time DTP was first proposed, Lillicrap et al. (2016) demonstrated that artificial neural networks can learn using so-called feedback connections that are inspired by the biological feedback connections in the brain, and did so by using fixed random weight matrices in place of the weight transpose when calculating each learning signal during the backwards pass. This approach, referred to as feedback alignment (FA), was claimed by the authors to be more biologically plausible than backpropagation as it addressed the implausibility of weight transportation in biological neural networks. FA and its derivatives would be further evaluated by Bartunov et al. (2018) and be shown to be very limited in comparison to the backpropagation algorithm on difficult tasks such as ImageNet. However, Moskovitz et al. (2019) would concurrently propose their own variations of the feedback alignment algorithm and make great strides in bringing these biologically motivated algorithms closer to the performance of backpropagation in deep convolutional neural networks. Following this initial work on feedback alignment, Nøkland (2016) proposed an alternative approach that connected each layer directly to the error through a fixed random weight matrix in the backward path. Called direct feedback alignment (DFA), this contribution was significant as it leveraged feedback alignment to enable backwards unlocking meaning that during training the gradients for each layer can be calculated in parallel. Unfortunately, just as the original feedback alignment method, direct feedback alignment has difficulty scaling to more difficult problems and training convolutional layers. In a follow up paper on DFA, Launay et al. (2019) showed that the approach simply failed to train convolutional layers. Later, Han & Yoo (2019) showed that VGG-16 could be trained with DFA if only the full connected layers are trained with DFA while the convolutional layers are trained with backpropagation, and Han et al. (2020) later showed that by only having direct connections to specific layers better performance in accuracy over DFA while training convolutional networks on the CIFAR10 dataset could be made. Despite this shortcoming, DFA shows fairly strong performance on various NLP tasks as shown by (Launay et al., 2020), and been used to enable higher power efficiency in SOC design (Han et al., 2019). Other follow up works to DFA helped to reduce the additional memory costs of DFA(Han et al., 2019; Crafton et al., 2019), and Frenkel et al. (2019) even showed that propagating targets in place of the gradient at the output can be just as effective. Further recLRA, proposed by Ororbia et al. (2020), showed strong performance on the ResNet architectures with its own biologically inspired derivation of DFA that, similarly to our proposed approach, updates the backward feedback connections, but this performance was achieved by the more practical method of only have direct feedback connections to some layers. More recently, Akrout et al. (2019) and Kunin et al. (2020) have shown that credit assignment approaches similar to FA can scale to larger problems by training the backward weights and even come close to matching the performance of backpropagation on the ImageNet classification task. Akrout et al. (2019) proposed weight mirroring (WM) which trained the backward weights to mirror their forward counterparts using the transposing rule and proposed another method, referred to as Kolen-Pollack learning, based on the research of Kolen & Pollack (1994), that updates the

backward matrices with the same gradient as the forward weights and uses weight decay on both the forward and backward matrices to encourage symmetry between the two. As a follow up to this, Kunin et al. (2020) proposed a set of regularization primitives with which to update the backward weights and combined these primitives into various configurations that showed improved performance and stability over WM. One such configuration, information alignment (IA), uses only local learning rules that follow what the authors consider key biological constraints to train various deep ResNet architectures on the ImageNet data set.

Since the introduction of DFA, a number of notable works have proposed their own unique approaches for enabling the parallelization of the forward and backward passes during training. Among these approaches, one of the most notable advances was made by Jaderberg et al. (2017) in their work on decoupled neural interfaces (DNI) which trains modules at each layer to predict synthetic gradients given the layer's input and some context information, such as the global target, as the input into the module, and then waiting for the true gradients in order to update each module's parameters. Nøkland & Eidnes (2019) used sub-networks at each layer to produce local loss functions with which to update the weights of the primary network. They also showed that a backprop free version of this approach can train with only a minor loss in accuracy. More recently, based on the earlier greedy layer-wise learning (DGL) method(Bengio et al., 2007), Belilovsky et al. (2019) proposed synchronous and asynchronous variants of DGL for addressing the issues of update locking and forward locking, and showed that these variants can match the performance of DGL and are capable of training deep convolutional neural networks on the ImageNet classification task. In the same vein as prior work which decoupled the backpropagation algorithm using delayed gradients(Huo et al., 2018), Xu et al. (2020) also recently introduced another novel training method, diversely stale parameters (DSP), which achieved comparable results to backpropagation on the ImageNet task.

## 2 APPROACH

### 2.1 BACKPROPAGATION AND DIRECT FEEDBACK ALIGNMENT

The back-propagation algorithm has long proven to be a robust, well performing approach for credit assignment in artificial neural networks. However, in recent years, there has been a search for alternatives that overcome the constraints of backwards and forwards locking which are found in backpropagation. One such alternative is direct feedback alignment (DFA), a bio-inspired credit assignment algorithm that enables backwards unlocking and the foundational approach for the method we are proposing. For a simple comparison of BP to DFA and our approach, we define a simple feed-forward neural network with $N$ layers, ignoring the biases. Then the learning signal $\delta_k$ for the output layer $k$ and the learning signal $\delta_\ell$ for some layer $\ell < k$ as prescribed by the backpropagation algorithm, where $a_\ell$ are our activations and $f'$ the derivative of some non-linearity such as a sigmoid function, are as follows.

$$\delta_k = error \odot f'(a_k). \tag{1}$$

$$\delta_\ell = \delta_\ell \cdot W_{\ell+1}^T \odot f'(a_\ell) \tag{2}$$

In the equations above and from here on out, $\cdot$ is the dot product operation between two matrices, and $\odot$ is the Hadamard product, or also known as the element-wise product. In DFA, $B_\ell$ is a fixed random weight matrix that projects the gradient $\delta_k$ at the output of a network to the output of layer $\ell - 1$. Thus, the learning signal $\delta_\ell$ for any layer $\ell$ upstream of output layer can be computed in the following way where $a_\ell$ are the activations and $f'$ the derivative of the non-linearity used at layer $\ell$.

$$\delta_\ell = \delta_k \cdot B_{\ell+1} \odot f'(a_\ell) \tag{3}$$

Then, every other aspect of the network is calculated in the same way as backpropagation. Thus, the weights $W_\ell$ at layer $\ell$ would be updated in the following way, where $\eta_W$ is the learning rate for the forward parameters, just as it is with backpropagation.

$$\nabla W_\ell = -\eta_W \delta_\ell \cdot a_{\ell-1}^T. \tag{4}$$

Because the backward weights are fixed, it is thought that the forward weights are aligning themselves with the backward weights such that the backward weights become useful for providing

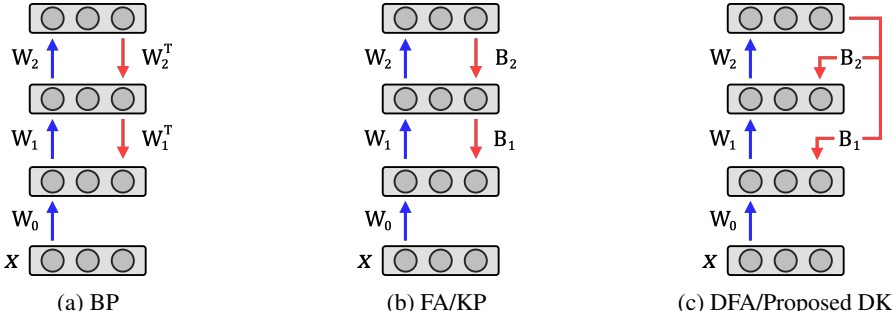

(a) BP  (b) FA/KP  (c) DFA/Proposed DKP

Figure 1: From left to right, the three figures above depict the forward(━) and backward(━) paths when training with backpropagation(a), feedback alignment or Kolen-Pollack learning(b), and direct feedback alignment or direct Kolen-Pollack learning (c) respectively.

meaningful updates to the network during training. However, Launay et al. (2019) showed that DFA, while showing results that nearly match backpropagation on the MNIST dataset with fully connected (FC) layers, fails to scale to harder problems such as CIFAR100 and shows significantly worse performance while training convolutional neural networks.

## 2.2 KOLEN-POLLACK LEARNING

For the purpose of overcoming the weight transportation problem, both Kolen-Pollack learning and the method it is based on, feedback alignment (FA), use a backward matrix at each layer, which shares the same dimensions as the transpose of the corresponding forward matrix, that takes the place of the weight transpose while calculating the learning signal at the upstream layer. While the backward path consists of fixed random weight matrices in feedback alignment, the backward matrices in Kolen-Pollack learning are updated with the transpose of the weight updates made to their forward counterparts, and weight decay is used on both the forward and backward weight matrices in Kolen-Pollack learning. Following our prior notation, the learning signal $\delta_\ell$ for some layer $\ell < k$ as prescribed by Kolen-Pollack learning and feedback alignment are as follows:

$$\delta_\ell = \delta_\ell \cdot B_{\ell+1} \odot f'(a_\ell). \tag{5}$$

For Kolen-Pollack learning, the update directions for $B_\ell$ and $W_\ell$ with weight decay, where $\lambda$ is a number between 0 and 1, are as follows:

$$\nabla B_\ell = -\eta_B \delta_\ell^T \cdot a_{\ell-1} - \lambda B_\ell, \quad \nabla W_\ell = -\eta_W \delta_\ell \cdot a_{\ell-1}^T - \lambda W_\ell. \tag{6}$$

In later sections we will discuss the learning dynamic of KP and how it applies to our proposed method.

## 2.3 DIRECT KOLEN-POLLACK

To address these issues with DFA, we propose a method for updating the backward matrices inspired by the work of Akrout et al. (2019) and Kolen & Pollack (1994) which we call direct Kolen-Pollack (DKP) learning. For DKP, the prior rules for DFA as stated in the previous section remain the same, however $B_\ell$ is no longer a fixed matrix. Rather, we will adjust the backward matrices after each batch using the following update rule where $\eta_B$ is the learning rate for the backward parameters and $\delta_k$ is the learning signal at the output layer $k$.

$$\nabla B_\ell = -\eta_B \delta_k^T \cdot a_{\ell-1}. \tag{7}$$

Similarly to the Kolen-Pollack approach used by Akrout et al. (2019), we also use weight decay on both the forward and backward parameters during training. Thus, the update directions for $B_\ell$ and $W_\ell$ with weight decay would be as follows where the hyper-parameter $\lambda$ is ranged between 0 and 1.

$$\nabla B_\ell = -\eta_B \delta_k^T \cdot a_{\ell-1} - \lambda B_\ell, \quad \nabla W_\ell = -\eta_W \delta_\ell \cdot a_{\ell-1}^T - \lambda W_\ell. \tag{8}$$

In our experiments, we found that weight decay on both the forward and backward matrices was crucial for maintaining the stability of the network during training, but we also try to substantiate this mathematically in the following sections.

While clear parallels exist between DKP and KP, the underlying learning dynamics between the two approaches are different as we will discuss in section 2.5 though we assert that the end result of these approaches are the same. This result being that the forward and backward connections align to estimate the same function.

## 2.4 KOLEN-POLLACK LEARNING AND CONVERGENCE THROUGH WEIGHT DECAY

Kolen-Pollack learning works on the principal of convergence through weight decay. Akrout et al. (2019) show that two synapses receiving the same arbitrary updates with an equal amount of weight decay will eventually converge on the same value. Before showing how this may apply to direct Kolen-Pollack learning, let us first reiterate how two synapses, and by extension two weight matrices that share the same dimensions, will converge in Kolen-Pollack learning.

We first consider a pair of discrete-valued forward and backward weight matrices, $W$ and $B$, that share the same dimensions such that all $i, j$ element pairs $W_{i,j}$ and $B_{i,j}$ are different at time step $t = 0$. Then, at every time step $t$ we update each pair of elements $W_{i,j}(t)$ and $B_{i,j}(t)$ with some arbitrary adjustment value $A_{i,j}(t)$ along with an equal amount of weight decay on both elements where $0 < \lambda < 1$. Thus, at time step $t$ the updates made to $W_{i,j}(t)$ and $B_{i,j}(t)$ are given as follows:

$$\nabla W_{i,j}(t) = A_{i,j}(t) - \lambda W_{i,j}(t), \;\; \nabla B_{i,j}(t) = A_{i,j}(t) - \lambda B_{i,j}(t). \tag{9}$$

Then

$$W_{i,j}(t+1) = W_{i,j}(t) + \nabla W_{i,j}(t), \;\; B_{i,j}(t+1) = B_{i,j}(t) + \nabla B_{i,j}(t). \tag{10}$$

For the sake of algebraically showing why the forward and backward weight matrices in Kolen-Pollack learning come to mirror one another, let us now consider the difference between $W_{i,j}(t+1)$ and $B_{i,j}(t+1)$ as

$$\begin{aligned}
W_{i,j}(t+1) - B_{i,j}(t+1) &= [W_{i,j}(t) + \nabla W_{i,j}(t)] - [B_{i,j}(t) + \nabla B_{i,j}(t)] \\
&= W_{i,j}(t) + [A_{i,j}(t) - \lambda W_{i,j}(t)] - B_{i,j}(t) - [A_{i,j}(t) - \lambda B_{i,j}(t)] \\
&= (1 - \lambda)[W_{i,j}(t) - B_{i,j}(t)] \\
&= (1 - \lambda)^{t+1}[W_{i,j}(0) - B_{i,j}(0)].
\end{aligned} \tag{11}$$

Since $0 < \lambda < 1$, we also find that $0 < 1 - \lambda < 1$. Thus, $\lim_{t \to \infty}(1 - \lambda)^{t+1} = 0$, and by extension $\lim_{t \to \infty}(1 - \lambda)^{t+1}[W_{i,j}(0) - B_{i,j}(0)] = 0$ since $W_{i,j}$ and $B_{i,j}$ are initialized as discrete values. Therefore as $t$ increases in value, the difference between $W_{i,j}(t)$ and $B_{i,j}(t)$ will approach zero. Since this is true for any pair of corresponding $i, j$ elements between $W$ and $B$, it follows that the same is true for the two weight matrices themselves, and thus, the following also should hold:

$$\sum_{i,j} \lim_{t \to \infty} [W_{i,j}(t) - B_{i,j}(t)] = 0 \tag{12}$$

Therefore, after enough time steps, $W$ and $B$ will come to approximately equal to each other. Note that while training a neural network with Kolen-Pollack learning, the updates made to the forward and backward weight matrices at each layer are not arbitrary. Thus, each pair of forward and backward matrices will come to equal each other while also minimizing the global error of the network. Because the forward and backward weight matrices do eventually come to approximately mirror one another, $i.e. B_\ell \simeq W_\ell^T$, we expect that the gradients made by Kolen-Pollack learning will eventually closely align with the gradients made by the backpropagation algorithm.

## 2.5 ALIGNMENT IN DIRECT KOLEN-POLLACK LEARNING

In direct Kolen-Pollack (DKP) learning, it is not immediately clear how the forward and backward paths are aligning since the forward and backward weight matrices at each layer do not share the

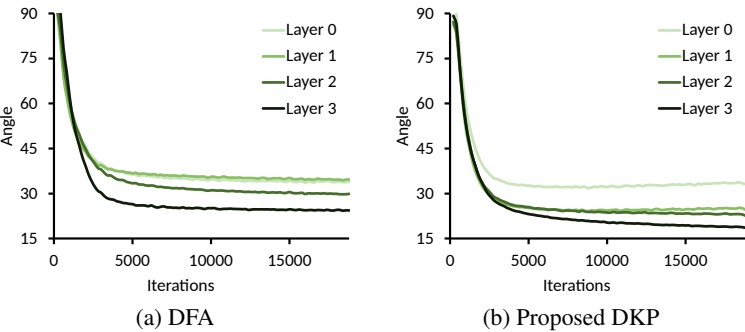

(a) DFA            (b) Proposed DKP

Figure 2: The cosine alignment angle between $a_{\ell-1} \cdot B_\ell^T$ and $h_k$ in degrees for each backwards matrix starting from the first layer(—) to the last layer before the output(—) for a duration of about 18,000 iterations. *In these results, lower angles equate to better alignment with the backpropagation algorithm.* The value of $\lambda$ used is $10^{-6}$. Note that the scales of y-axis in both figures are the same.

same dimensions. The only exception is the second to last layer $(k-1)$ where the forward and backward weight matrices do share the same dimensions, and thus, we expect this layer to behave as it would in Kolen-Pollack learning. In Kolen-Pollack learning, we can say that some backward matrix $B_\ell$ is converging on the function $f_\ell$ that maps the input $a_{\ell-1}$ of layer $\ell$ to the preactivation hidden state $h_\ell$ i.e $f_\ell \colon a_{\ell-1} \mapsto h_\ell$. As we showed in the previous section, this method of convergence works with Kolen-Pollack learning because $f_\ell$ is a linear mapping that uses the same dimensions as $B_\ell^T$. However, with direct Kolen-Pollack learning the function $f_\ell$ is no longer a linear function for all layers $\ell < k - 1$ and instead is the nonlinear function $f_\ell \colon a_{\ell-1} \mapsto h_k$ which is comprised of layers $\ell$ through $k$. Due to the highly nonlinear nature of $f_\ell$ alone it would not be reasonably possible for any direct linear function in the backward path to converge with this function.

However, we might expect these direct feedback connections in DKP to converge with a linear function that estimates the nonlinear forward connections. That is to say $B_\ell$ converges on some linear function $g_\ell$ that estimates the nonlinear function $f_\ell \colon a_{\ell-1} \mapsto h_k$. To substantiate these claims we experimentally show that as a network trains with DKP the cosine alignment angle between the preactivation output of the network $h_k$, to which all functions $f_\ell$ map to, and the dot product of the activations $a_\ell$ at any layer $\ell < k - 1$ and the transpose of the corresponding direct backward weight matrix $B_\ell$ will closely align after a sufficient number of training steps has passed. The network we will be using to examine this property of DKP is a five layer fully connected neural network trained on MNIST with cross entropy used for the loss function. After each hidden layer, batch normalization is used and followed with a rectified linear unit activation function.

In figure 2b, we see that with DKP the cosine alignment angle between $a_{\ell-1} \cdot B_\ell^T$ for each layer $\ell < k$ and the output of the network $h_k$ is much lower than it is with DFA in figure 2a with the exception of the angle measured at the first layer which does not seem to improve much from DFA to DKP. Thus, we see that both DFA and DKP seem to exhibit this behavior of estimating the nonlinear forward path with some linear function, however DKP clearly does this to greater effect. We also notice that generally the direct feedback connections further downstream align much more closely with the forward connections than those further upstream. This result likely occurs because the upstream feedback connections are converging to some linear function that has to estimate a forward function that is much more complex and highly nonlinear relative to the downstream functions.

## 3 EXPERIMENTS

All of our experiments were coded with Python using the PyTorch library. Code will be available on GitHub soon.

### 3.1 EXPERIMENTAL SET-UP

In our experiments, using batch normalization was necessary to gain stable training for both DFA and DKP, and both benefit significantly from its usage depending on the weight initialization method.

This was especially an important for DKP as without batch normalization we would often run into an issue of exploding gradients. Additionally, for DFA and DKP we use Kaiming weight initialization(He et al., 2015) on the backward weight matrices as suggested by Launay et al. (2019).

We also found that the optimal hyperparameters and optimizers for the backward weight matrices in DKP seem to vary a some from one architecture to the next. For our first set of experiments with convolutional networks we employ a smaller network with only two convolutional layers to train on the CIFAR10 and Fashion-MNIST datasets, and for this network, training on CIFAR10, we found that the best optimizer for the backward weights was simply the same as that used on the forward weights, SGD with Nesterov acceleration and momentum at 0.9, with the only difference being a reduced learning rate. On the forward parameters a learning rate of $10^{-2}$ was used, and on the backward parameters a learning rate of $10^{-4}$ was instead used. Both forward and backward parameters used weight decay with $\lambda = 10^{-6}$. Additionally, we used a step-wise learning rate decay on the backward parameters with $\gamma = 0.85$ which further improved the training stability. However, in our second experiment training the AlexNet architecture on the CIFAR100 dataset and TinyImageNet200 dataset, neither of these previous strategies involving the optimizer and learning rate scheduler were optimal. For training with AlexNet on CIFAR100, the Adam optimizer (Kingma & Ba, 2014) was used on the backward weight matrices with a learning rate of $5 \times 10^{-4}$, while SGD with Nesterov acceleration, a learning rate of $0.01$, and a momentum of $0.9$ was again used on the forward weight matrices. In the case of the AlexNet tests, both the forward and backward parameters used weight decay where $\lambda = 10^{-4}$. The only difference moving to the TinyImageNet200 dataset was a slight decrease of the forward learning rate to $5 \times 10^{-3}$ for all approaches except the backpropagation tests.

We believe that the differences in what was optimal for training with DKP between the two networks were the result of the increased difficulty in the problems being solved and potentially the size of the networks themselves. However, the weight initializations, hyperparameters, and optimizers used in our experiments were not found through a rigorous search, and thus further improvements to the performance of these learning algorithms could potentially be found. We also note that in all of our experiments, the learning rates seen are about the highest values that one would want to use before running the risk of exploding gradients, and that for all methods it is perfectly acceptable to use lower values. However there is a slight exception when training with KP and DKP: if the learning rate on the backward weights are too low, then these methods begin to behave as their static counterparts FA and DFA respectively which leads to reduced performance. Because the change from DFA to DKP is analogous to the change from FA to KP, we will be including FA and KP in our experiments for comparison to their direct counterparts. These additional experiments will help to illustrate the difficulty that arises from training with direct feedback connections and that the application of Kolen-Pollack learning alone is not enough to overcome their shortcomings.

## 3.2 TRAINING CONVOLUTIONAL NETWORKS

Because DFA has a difficult time training convolutional layers, achieving a more meaningful result with direct feedback connections in these deep convolutional networks is an important step and the primary improvement DKP makes over its predecessor. To demonstrate DKP's capability in training convolutional layers we not only compare it to backpropagation and DFA but also we test each network's performance with the training on the convolutional layers frozen such that only the fully connected layers are trained with backpropagation; anything that performs less than or equal to this in terms of accuracy or loss is likely not training the convolutional layers effectively. We also compare it to direct random target projection(DRTP) (Frenkel et al., 2019) which projects the one hot encoding of the target directly to each layer. The loss function used in all of our experiments is the cross entropy loss. All reported numbers are an average of 10 trials.

For our first experiment, we train on the CIFAR10 dataset for $50$ epochs using a network that consists of just two convolutional layers followed by two fully connected layers, the second being the output layer. The results in Table 1 show that DFA does considerably better than BP with the convolutional layers frozen, and that DKP performs even better than DFA. Of course, we still see that BP is out performing both DFA and DKP by a much larger margin.

In our second experiment, we train on the CIFAR100 and TinyImageNet 200 datasets using the AlexNet architecture for $90$ epochs, and again, batch normalization is used before the activation

Table 1: 2-Conv. Layer network results on CIFAR10 and Fashion-MNIST with cross entropy loss. Serial/Parallel indicates serial training and parallel training scheme during the backwards pass. Parallel+ indicates that the method is capable of update unlocking.

| Dataset | Method | Serial/Parallel | Inference Accuracy | Train Accuracy |
|---|---|---|---|---|
| Fashion-MNIST | BP (FC Only) | Serial | $91.33\% \pm 0.18$ | $99.97\% \pm 0.02$ |
| | BP (Upperbound) | Serial | $92.18\% \pm 0.13$ | $100.00\% \pm 0.00$ |
| | KP | Serial | $91.25\% \pm 0.18$ | $99.47\% \pm 0.14$ |
| | FA | Serial | $91.12\% \pm 0.39$ | $99.41\% \pm 0.26$ |
| | DRTP | Parallel+ | $89.58\% \pm 0.05$ | $94.86\% \pm 0.02$ |
| | DFA | Parallel | $91.54\% \pm 0.14$ | $99.88\% \pm 0.05$ |
| | **DKP (Ours)** | Parallel | $\mathbf{91.66\% \pm 0.27}$ | $\mathbf{99.89\% \pm 0.06}$ |
| CIFAR10 | BP (FC Only) | Serial | $60.01\% \pm 1.32$ | $99.34\% \pm 0.63$ |
| | BP (Upperbound) | Serial | $70.70\% \pm 0.96$ | $99.82\% \pm 0.49$ |
| | KP | Serial | $70.08\% \pm 0.37$ | $99.98\% \pm 0.01$ |
| | FA | Serial | $60.45\% \pm 1.13$ | $95.36\% \pm 1.46$ |
| | DRTP | Parallel+ | $55.32\% \pm 6.14$ | $72.90\% \pm 0.08$ |
| | DFA | Parallel | $62.70\% \pm 0.36$ | $97.72\% \pm 1.24$ |
| | **DKP (Ours)** | Parallel | $\mathbf{64.69\% \pm 0.72}$ | $\mathbf{99.09\% \pm 0.29}$ |

Table 2: AlexNet results on CIFAR100 and TinyImageNet200 with cross entropy loss. Serial/Parallel indicates serial training and parallel training scheme during the backwards pass. Parallel+ indicates that the method is capable of update unlocking.

| Dataset | Method | Serial/Parallel | Inference Accuracy | Train Accuracy |
|---|---|---|---|---|
| CIFAR100 | BP (FC Only) | Serial | $47.72\% \pm 0.73$ | $39.03\% \pm 0.32$ |
| | BP (Upperbound) | Serial | $65.88\% \pm 1.02$ | $64.09\% \pm 0.39$ |
| | KP | Serial | $66.78\% \pm 0.47$ | $67.70\% \pm 1.88$ |
| | FA | Serial | $19.49\% \pm 0.97$ | $12.90\% \pm 0.80$ |
| | DRTP | Parallel+ | $5.84\% \pm 0.65$ | $5.49\% \pm 0.19$ |
| | DFA | Parallel | $48.03\% \pm 0.61$ | $35.18\% \pm 0.43$ |
| | **DKP (Ours)** | Parallel | $\mathbf{52.62\% \pm 0.48}$ | $\mathbf{45.17\% \pm 0.43}$ |
| TinyImageNet200 | BP (FC Only) | Serial | $29.78\% \pm 0.45$ | $24.89\% \pm 0.24$ |
| | BP (Upperbound) | Serial | $49.53\% \pm 0.61$ | $49.25\% \pm 0.31$ |
| | KP | Serial | $51.36\% \pm 1.50$ | $60.44\% \pm 1.94$ |
| | FA | Serial | $9.98\% \pm 1.44$ | $7.12\% \pm 0.83$ |
| | DRTP | Parallel+ | $2.86\% \pm 0.59$ | $2.86\% \pm 0.26$ |
| | DFA | Parallel | $32.116\% \pm 0.66$ | $24.75\% \pm 0.36$ |
| | **DKP (Ours)** | Parallel | $\mathbf{35.78\% \pm 1.92}$ | $\mathbf{35.97\% \pm 0.37}$ |

| Layer No. | Layer Configuration |
|---|---|
| 0 | Conv(Channels: 3-32, Kernel: 3, Padding: 1), BN., ReLU |
| 1 | Conv(Channels: 32-32, Kernel: 3, Padding: 1), BN, ReLU |
| 2 | Max Pool(Kernel: 2), Linear(6272, 128), BN, ReLU |
| 3 | Linear(128, 10) |

Table 3: 2-Conv. Layer network architecture used in the first set of experiments. BN stands for batch normalization.

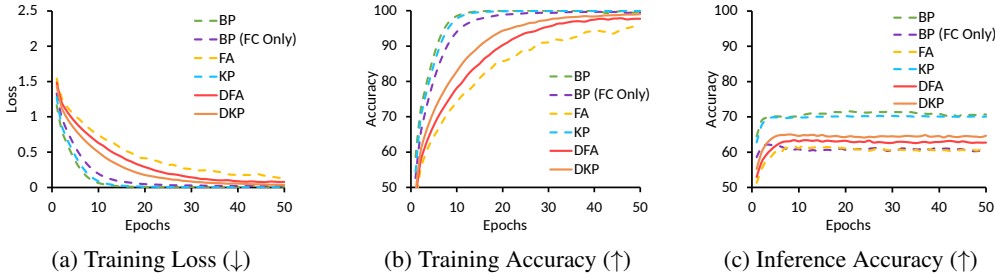

(a) Training Loss (↓)           (b) Training Accuracy (↑)           (c) Inference Accuracy (↑)

Figure 3: From left, the training loss, training accuracy, and test accuracy of the CIFAR10 experiments using 2-layer CNN. ↓ indicates lower the better, ↑ indicates higher the better. Dashed lines represent serial training and solid lines represent parallelizable training during the backwards pass.

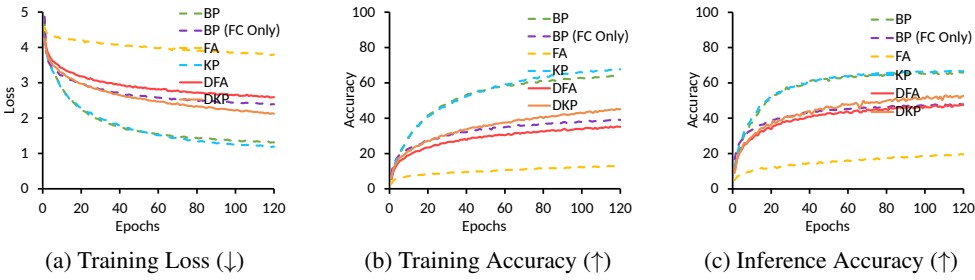

(a) Training Loss (↓)           (b) Training Accuracy (↑)           (c) Inference Accuracy (↑)

Figure 4: From left, training loss, training accuracy, and test accuracy on the CIFAR100 experiments using the AlexNet architecture. ↓ indicates lower the better, ↑ indicates higher the better. Dashed lines represent serial training and solid lines represent parallelizable training during the backwards pass.

functions on all layers except the output layer. For CIFAR100, while we worked to get as close to a direct comparison as we could between BP, DFA, and DKP in terms of hyperparameters and weight initializations, we did have to lower the learning rate from 0.1 to 0.01 on the forward parameters with DFA and DKP to achieve more stable training; lowering these values for the backpropagation tests resulted in slightly lower accuracies. Similarly, the TinyImageNet200 results required slightly lower learning rates. The results in table 2 are mostly consistent with what is seen in the CIFAR10 test. DKP, while showing a solid jump in performance over DFA, still fails to match the performance of BP, and DFA only performs marginally better than BP when only the fully connected layers trained.

We see that as the contribution to the results in accuracy made by the fully connected layers diminishes, so too does the results of DFA. Also, we see that as the networks become more reliant on the convolutional layers to perform well, the gap between DFA and DKP in terms of accuracy widens in favor of our approach. We also note that the direct random target projection(DRTP) approach suffers terribly from the usage of the cross entropy loss despite performing well on networks similar to our experiments in table 1 when using mean squared error. So another benefit of DKP is that it is compatible with a more robust loss function, cross entropy.

## 4 CONCLUSION

Direct feedback alignment (DFA) enables the parallelization of the backwards pass, called backwards unlocking, and has shown promising results in NLP tasks(Launay et al., 2020). Despite these clear advantages, DFA fails to effectively train deep convolutional networks on difficult image classifications tasks. We propose direct Kolen-Pollack (DKP) learning by incorporating Kolen-Pollack learning into DFA to more effectively train deep convolutional neural networks with direct feedback connections by updating the backward weight matrices. We empirically show that our approach produces gradients that better align with the backpropagation algorithm. We further show that DKP outperforms DFA while training convolutional neural networks on the Fashion-MNIST, CIFAR10, CIFAR100, and TinyImageNet200 datasets.

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
