# OpenReview forum: "Learning the Connections in Direct Feedback Alignment"
_ICLR.cc/2021/Conference — Reject_

### Official Review · AnonReviewer4 · 2020-10-23
**This paper proposed a direct Kolen-Pollack (DKP) method which updates the weight of the backward pass of DFA and achieves better performance.**

**Rating:** 5
**Confidence:** 3

**Review:**

Strength:
As mentioned in the paper, although DFA is more biologically plausible, it does not work well for deep networks and CNNs. This work proposed a possible improvement for this problem.

Weakness:
(1) My main concern is the novelty of this paper. To my understanding, the DKP and the convergence of weight decaying are first proposed in (Akrout et al., 2019). However, updating the backward matrix suffers from the problem of different dimensions. I think the main contribution of this work is the experimental demonstration of the lower angle achieved by DKP. Therefore, I hope the authors can make it clear the main contributions of this work. It is also necessary to comment on how the approach in this paper is different from (Akrout et al., 2019).

(2) In the experiments, the authors only compare the DKP with the DFA and BP. It is also necessary to compare the performance with other references. For example, how much the performance can be improved by the proposed method compared to the methods introduced in section 1.1.

(3) In addition, the network sizes are not clearly mentioned in the experiments. It is difficult for readers to make a judgment about the effectiveness of this work.

---

> ### Author Response · Authors · 2020-11-20
> **Thank you for sharing your concerns and valuable feedback!**
>
> Thank you for taking the time to write your review of our submission. We appreciate and value the feedback you have given us. We hope that we can adequately address your concerns. We would also like to apologize for the delayed response and ask for the reviewer’s understanding of this matter. Thank you!
>
> Based on the feedback of the other reviewers we have updated our submission to include a more difficult dataset.
>
> 1.
> We acknowledge that we did not clearly state all of our contributions and have updated our introduction accordingly. Thank you for pointing out this shortcoming of our submission.
>
> A direct variant of Kolen-Pollack learning, what we propose in our submission and call DKP, was not proposed in (Akrout et al., 2019), and as we stated in our paper, “Akrout et al. (2019) show that two synapses receiving the same arbitrary updates with an equal amount of weight decay will eventually converge on the same value… let us first reiterate how two synapses, and by extension two weight matrices that share the same dimensions, will converge in Kolen-Pollack learning.” We make no claim to be the first to present such an algebraic argument for Kolen-Pollack learning and properly state that this was first done by (Akrout et al., 2019). Our contribution in section 2.4(changed from 2.3) is simply a more detailed explanation than was given by (Akrout et al., 2019) as KP was not the focus of their paper but is an important aspect of ours.
>
> In section 2.5, we also provide a discussion on the differences in the learning dynamics present in KP and DKP and have updated our submission to further clarify these differences.
>
> The novelty in this work is not just in the application of KP to DFA and the issues regarding alignment, but also in the recommended training procedures for training CNNs with direct feedback connections. We state that “using batch normalization was necessary to gain stable training for both DFA and DKP, and both benefit significantly from its usage… without batch normalization we would often run into an issue of exploding gradients”. This point is not highlighted in other works on DFA and some works (Launay et. al 2019) suggest that BN may be detrimental to the performance of DFA in convolutional neural networks but this is likely due to their weight normalization techniques. We also suggest effective optimizers for training direct backward weight matrices. Additionally, we provide important insight into training with DFA. No other works show meaningful performance of the level that we show with DFA on any CNN as deep as AlexNet, and no other works train DFA with datasets more challenging than the CIFAR10 dataset while still showing meaningful results. The contributions to the usage of DFA itself were an understated aspect of our submission and we have updated our submission to address this.
>
> We have also updated our descriptions of KP and DKP to better clarify the differences present in their learning dynamics.
>
> 2.
> Based on the feedback of the reviewers we will be updating our experiments with an additional relevant work, direct random target projection(DRTP) (Frenkel et al. 2019), but we would also like to point out that the majority of the related works are not directly comparable to DFA or our approach, and that some of the more potentially relevant works do not build their approaches for the purpose of connecting feedback connections/auxiliary networks to all layers as the motivations of those papers are different from ours. A primary motivation for approaches such as DFA and DKP is their potential to be used for edge devices, IoT, etc due to their low computational cost. This is why we have decided to include DRTP in our experiments. We have updated our introduction to give more detail on this motivation.
>
> 3.
> We appreciate this feedback as we want our work to be easily read and understood. We value this quality in academic writing. Based on the feedback of all of the reviewers we have added a chart to clearly state our network architecture for the first experiment and have updated our experimental set-up section with more detail regarding loss functions and hyperparameter settings.
>
> Thank you for stating your concerns and providing your input! We are grateful for your clear and concise feedback.
>
>
> (Launay et. al 2019, Principled Training of Neural Networks with Direct Feedback Alignment)
> (Akrout et al., 2019, Deep Learning without Weight Transport)
> (Frenkel et al. 2019, Learning without feedback: Direct random target projection as a feedback-alignment algorithm with layerwise feedforward training)

---

### Official Review · AnonReviewer3 · 2020-10-28
**iterative work, need more clarification**

**Rating:** 5
**Confidence:** 4

**Review:**

## Second Review

I thank authors for taking time and answering my queries. However current manuscript fails to point out key difference between Akrout 19 kolen-pollack method and DKP (proposed method). Combining FA with DKP does not add sufficient novelty. As pointed out by other reviewers, paper should highlight key differences and reasoning for such combination. I am happy to see additional results with DRTP, however it is also important to test your approach based on there methodology.   It is difficult to gauge the significance of your approach, since training protocol varies a lot. I would request authors to add more baselines and training protocols  (future submission) to show that your method is robust and can also train deeper CNNs models. Current submission missed out on many key aspects, despite having promising direction. I hope our reviews help you in strengthening this promising work.
## Summary

This work proposes an approach to update feedback weights in DFA using modification of kolen-pollack method, which helps in training deep CNN network.

## First Review
Citation missing for key work on assessing the scalability of bio-inspired approaches and highlighting key limitations [Bartunov 18], variants of DFA [ Moskovitz 18, Frenkel 19]  and recently an approach similar to DFA with target projection known as LRA (also has similarity with Direct Kolen-Pollack) showing promising performance on deep CNNs [ Ororbia & Mali 2020].

The update rule used in recursive-LRA is similar to what proposed in this paper
Delta_b(update for feedback weights)  = learning rate *( teaching signal(delta_k) * post-activation from layer below (a_(l-1))

For LRA Delta_b(update for error weights)  = learning rate *( teaching signal(error_k) * post-activation from layer below (a_(l-1))

For weight mirroring [ Akrout 19] Delta_b(updates for feedback weights) = = learning rate *( teaching signal(delta_(l) * delta(updates) from layer above (delta_(l+1))

One can see we can derive chain rule formulation with certain assumption in which feedback matrix or error matrix acts like transpose of forward weights (rotated 180 degree).

As shown in Feedback alignment(lillicrap 16) the updates for FA and LRA lie with 90-degree w.r.t BP. One can provide such plots to show how far way are your updates w.r.t. BP and other bio-inspired approach.

“We also found that the optimal hyperparameters and optimizers for the backward weight matrices in DKP seem to vary greatly from one network to the next”
Can you provide more detail about your experimental setup? What are the range of hyper-parameters and how does DKP perform w.r.t BP and other variants? It is well known that DFA in its vanilla form suffer whenever tested with deep networks on challenging benchmarks such as imagenet (akrout 19, Bartunov 18). As shown by Moskovitz 18 and Crafton 19, integrating BP or making feedback weights close to forward weights helps in learning for complex benchmarks. So, what different does DKP offer? is it robust, speeds up the convergence, always stays consistent (robust against bad initialization). Current manuscript fails to highlight these points which could make current work stronger.

Do you constraint your feedback weights, if so how? If not, then how does model ensure that feedback weights are respecting forward neural activities and helping it to converge? Won’t feedback weights grow making discrepancy between forward and backward activities, thus slowing the convergence of the network?

Comparison against other variants of DFA
We would like to see detailed comparison w.r.t various variants or family of FA(Moskovitz 18, Frenkel 19] and LRA (since update rules are similar).





[Bartunov 18] Bartunov, S., Santoro, A., Richards, B., Marris, L., Hinton, G.E. and Lillicrap, T., 2018. Assessing the scalability of biologically-motivated deep learning algorithms and architectures. In Advances in Neural Information Processing Systems (pp. 9368-9378).

[Akrout 19] Akrout, M., Wilson, C., Humphreys, P., Lillicrap, T. and Tweed, D.B., 2019. Deep learning without weight transport. In Advances in neural information processing systems (pp. 976-984).

[Moskovitz 18] Moskovitz, T.H., Litwin-Kumar, A. and Abbott, L.F., 2018. Feedback alignment in deep convolutional networks. arXiv preprint arXiv:1812.06488.

[Frenkel 19] Frenkel, C., Lefebvre, M. and Bol, D., 2019. Learning without feedback: Direct random target projection as a feedback-alignment algorithm with layerwise feedforward training. arXiv preprint arXiv:1909.01311.

[Ororbia and Mali 20] Ororbia, A., Mali, A., Kifer, D. and Giles, C.L., 2020. Reducing the Computational Burden of Deep Learning with Recursive Local Representation Alignment. arXiv preprint arXiv:2002.03911.

---

> ### Author Response · Authors · 2020-11-20
> **Thank you for your detailed feedback and suggestions! We have updated our paper to address your concerns.**
>
> 1.
> Thank you for taking the time to clearly state your concerns and for pointing out key criticisms to help us improve our work. We would like to sincerely apologize for not being able to respond sooner and appreciate the reviewer’s understanding.
>
> 2.
> The citation for [Frenkel 19] is already included in the related works section. As for the other citations, it is an oversight that they were not at the very least included within the related works section and we have updated this section accordingly and added DRTP to our experiments (please note that the diminished performance of DRTP is due to the choice of loss function which is discussed in the new draft). Thank you! These works, while important and relevant to the broader topic of feedback alignment and parallelization, would not change the discussions had throughout our work. The paper on recLRA does not experiment with direct connections to all layers in a CNN and only to “blocks” or sections of a convolutional network. The focus of this work is on the case of direct feedback connections to all layers as this may be more useful for edge devices and the sort.
>
> 3.
> According to Algorithm 2 in [Ororbia & Mali 2020], the update rules for specifically direct connections in recLRA as written in the above notation would be as follows…
> Delta_b_l(update for error weights) = learning rate * d_(l-1)^T * teaching signal(error_k)  where d_(l-1) = b_l * teaching signal(error_k)
> As one can see, when training with recLRA, the update rules for a direct backward matrix b contains an element d that involves the dot product of b itself and does not include a_(l-1). DKP is much simpler and is shown to be more effective than prior works when direct connections are made to all layers in a CNN. [Ororbia & Mali 2020], while mentioning the possibility, do not conduct such experiments with CNNs as their work focuses on having error pathways from the network’s output connecting only to the output of some block of convolutional layers in a CNN.
>
> 4.
> We opted to measure angles that show how well the backward connections estimate the output of the forward connections as this measurement better displays our reasoning for why DKP works and provides insight into the learning dynamics of DFA not seen in prior works.
>
> 5.
> Based on the reviewer’s suggestion we will add a chart that more clearly states our network architecture. We have also updated the experimental set-up section with more detail. Thank you!
>
> 6.
> If the backward lr for DKP is too low, then performance suffers as it will behave like DFA. For the same reason, KP also has this problem. Other than this, as we state, “the weight initializations, hyperparameters, and optimizers used in our experiments were not found through a rigorous search." They were not over-tuned nor do they need to be. Both the forward and backward learning rates we selected are about the highest value one would want to use before running the risk of exploding gradients. Lower lr values for each approach are fine, except in the case with DKP and KP we discussed, but just simply lead to slower convergence. We have made the appropriate updates.
>
> 7.
> The scenario in which all layers receive direct feedback connections is what we focus on in our submission. And while for current PC hardware, and also from a software perspective, such implementations may not be practical, in the future it may be possible and useful for edge devices, IoT and SOC design. As we state in our related works section, DFA has “been used to enable higher power efficiency in SOC design (Han et al., 2019).” Our approach only has a minimal computational increase over DFA. DNI, DGL, and recLRA require more computations than DFA and DKP, and more importantly, these works give little or no consideration to directly connecting the output to all layers. In this context, we believe that the improvement shown by DKP is significant. We have updated our submission to clarify this.
>
> 8.
> In section 2.2 we state that using “weight decay on both the forward and backward matrices was crucial for maintaining the stability of the network”. Weight decay here also helps to ensure that our feedback weights do not grow out of control. No other restrictions are placed on the feedback weights. We have updated our manuscript.
>
> 9.
> We have added DRTP to our experiments. However, with considerations for time and a lack of code provided by the authors to check our implementation against, we will have to forego experiments for recLRA. Also, we did not include the variants of FA as the list is even longer than that provided by the reviewer and had to make length considerations. We agree that deeper discussions comparing FA and DFA would be a great contribution to the community. We have updated our submission to explain why FA and KP are included in the experiments.
>
> Again, we really appreciate your time, effort, fair criticism, and helpful attitude. We hope that our response has helped to address your concerns.

---

### Official Review · AnonReviewer2 · 2020-10-28
**DKP shows an interesting improvement over DFA for CNNs but results and analysis are slightly insufficient**

**Rating:** 6
**Confidence:** 3

**Review:**

Thanks for author(s) for their paper. I enjoyed reading it.

This paper introduces a new method for computing the backward updates of a neural network called Direct Kolen-Pollack learning (DKP). Similar to Direct Feedback Alignment (DFA), the aim of this work is to introduce a viable alternative to back-propagation (BP) that works in parallel while achieving similar performance. Parallelization benefits comes from the fact the the backward path is unlocked. To summarize, DKP shows an interesting improvement over DFA for CNNs but results and analysis are slightly insufficient.

Kolen-Pollack learning (KP) suggests “that updates the backward matrices with the same gradient as the forward weights and uses weight decay on both the forward and backward matrices to encourage symmetry between the two.” The idea in DKP is simple: replace error signal from each layer with the final layer’s error signal in KP. ($\delta_l \rightarrow \delta_k$ in Eq. 6 and 8)
Author(s) identified the performance gap between DFA and BP in CNNs as their motivation and tested their method on image classification on Fashion-MNIST, CIFAR10 and CIFAR100 datasets.

Notes:
1. The writing is easy to understand and clear. Authors are planning to release the code soon.
2. Please introduce matrix operations used in Eq. 1 and 2 somewhere.
3. In DKP backward matrices are no longer fixed as in DFA but rather are updated after each batch with their own update rule and learning rate. These updates are still parallelizable but an analysis and/or experimentation on the speed gain with these new updates are needed, specially comparing to BP and DFA. Similarly, in the related work it’s mentioned that “[previous works except DFA] currently show no tangible benefits over backpropagation as they all have larger memory requirements” but I don’t see any discussion on that in the paper.
4. Following prior works, weight decay on both the forward and backward parameters during training are used. Author(s) mention that weight decay is important to make this method work. As another contribution they provide mathematical justifications for its use. The result of this argument is that following this update rule after enough time, the forward and backward matrices will be close to each other. Please clarify following questions:
    4. i. This argument depends on $A_{i,j}$ being the same for backward and forward update. But comparing Eq. 8 and 9, doesn’t this require $\eta_B=\eta_W$?
    4. ii. This problem is worse with DKP when comparing Eq. 6 and 9 as ${\left(-\eta_B \delta_k^T . a_{l-1} \right)}_{i,j} \neq {\left(-\eta_W \delta_l . a_{l-1}^T \right)}_{i,j}$.
5. Description of Kolen-Pollack learning is minimal and a bit too late in the paper despite the proposed method is named after it. In my opinion it would’ve been better to explain it a bit more and earlier. For example the first few paragraphs of section 2.3 could move to earlier sections.
6. Figure 2: Value of $\lambda$ is missing. Also if I am not mistaken according to Eq. 11 the rate of convergence of the two should be proportional by $(1-\lambda)^t$. I am not certain that I see this trend in the graphs, specially layers 0 and 1 that seem to be plateauing. I understand that you have similar observation at the end of section 2. The explanation seems to be that higher layers are easier to be linearly approximated. However, my counterpoint is that Eq. 11 shows an exponential decay to zero and it does not depend on layer l. Could you comment on this please?
7. Despite the fact that KP does not make any assumptions about the structure and inductive biases of the network, DKP is proposed only for CNNs and image base classifications. Why shouldn’t DKP be used for MLPs, RNNs, etc.? I would really like to see its performance compare to BP and DFA for non-CNN structures.
8. In related works, author(s) mention prior works that DFA have a hard time with VGG-16 optimization but the experiments are done AlexNet. What is the reason for this mismatch. It would have been much easier to make a direct comparison with previous works.

---

> ### Author Response · Authors · 2020-11-20
> **Thank you for your detailed response! We have made changes to our paper based on your suggestions.**
>
> Thank you for your kind words! We are elated to hear that you enjoyed reading our submission and that our writing was clear and easily understandable. These attributes are important to us and we have made some adjustments to our paper based on the suggestions given to us by all of the reviewers. We are grateful for your detailed and well-organized review of our paper.  We know it must have taken a considerable amount of time and want to express our gratitude as your responses are incredibly valuable to us. We want to also just quickly apologize as we truly desired to respond sooner but some exceptional circumstances caused us a delay.
>
> To address the “slightly insufficient” results we have added an additional and more difficult dataset to our experimental results. We see that this is a common concern among the reviewers and wanted to address it the best we could with the time remaining.
>
> 2.
> We have fixed this in the new draft. Thank you!
>
> 3.
> For DFA and its variants, discussions on the exact benefits in terms of speed gains have been lacking in all prior literature. However, no current hardware can fully take advantage of the speed benefits provided by DFA for many modern convolutional architectures. Any discussion on the speed gains of DFA for such applications would be entirely theoretical until the hardware is available. This is also true for the core target hardware of this paper(edge devices, IoT, SOC \etc). Additionally, in the current landscape of research on the topic of DFA, the focus is often centered around improving the algorithm’s inference capability as without performance that matches backprop in a majority of applications the theoretical speed gains are a bit of a moot point.
>
> 4.
> The theoretical results for Kolen-Pollack learning as first described by [Arkout et. al 2019] would require that (a) the matrices are the same size, (b) that the updates to both matrices are the same, and (c) that the learning rates are the same as well. We found that lowering the learning rate on the backward weights can be helpful for preventing exploding gradients in both KP and DKP, and so our notation reflects this.
>
> The learning dynamics of DKP and KP produce the same result which is the backward pathways estimating the function that is the forward pathways, but the underlying learning dynamics are different. Unfortunately, proving this algebraically does not seem to be a straight-forward option as it was with KP, but the experimental results in Figure 2. do reflect this convergence of the forward and backward paths that do occur in KP. We have updated our draft to better clarify this.
>
> 5.
> We agree with the reviewer’s suggestion that would help to improve the logical flow of our paper. We have made this change in our new draft. Thank you!
>
> 6.
> Thank you for pointing this out, we have added the value of λ used for the experiments in figure 2.
> Under DKP the backward and forward weight matrices are only an estimate of each other and cannot ever be equal as a mere consequence of their dimensions and the non-linear qualities of the forward connections. This alone means that it simply is not possible for DKP to display this attribute of Kolen-Pollack learning. The best we could hope for is a noisy estimation of the learning dynamics present in KP. We have updated our draft to better convey this point.
>
> 7.
> It is well known that DFA and its variants train very well on fully connected networks and DKP is no exception. Though we did not state the inference results, we do include experiments with DKP training an ANN(Figure 2) for measuring alignment angles and show that DKP aligns better than DFA. As we stated in the related works, DFA has been “shown to perform reasonably well on a number of natural language processing tasks with recurrent neural networks and transformers by Launay et al. (2020)”, so it would be great to see how DKP holds up in these scenarios. However, we had to make considerations for the length of our paper.
>
> 8.
> Because DFA and its variants use fully connected backward connections to the output of all layers, the memory requirements can be quite large, especially for convolutional networks as one can imagine. Some works try to circumvent this issue by using a shared weight matrix that is referenced by the feedback connections in DFA. Unfortunately, this trick will not work with DKP, and due to hardware limitations, we were not able to include VGG16 in our experiments. It is for this reason we use AlexNet. We would argue that this is sufficient as it shows that DKP is an improvement, but that more research in the way of direct feedback connections is necessary before harder problems and architectures are to be considered anyhow.
>
> Once again, thank you for sharing your comments, suggestions, and concerns with us! All of your questions regarding the technical aspects of our work were very thorough. We hope that we properly addressed each of your points; it was a pleasure responding to each of them.

---

### Decision · Program_Chairs · 2021-01-07
**Final Decision**

**Decision:**

Reject

**Comment:**

This paper investigates an improvement to the direct feedback alignment (DFA) algorithm where the "backward weights" are learned instead of being fixed random matrices. The proposed approach essentially applies the technique of DFA to Kolen-Pollack learning. While reviewers found the paper reasonably clear and thought the experiments were acceptable, there were significant concerns about the novelty of the approach and the fact that the proposed approach was a straightforward combination of existing ideas. Further, the paper could have done a better job situating (and applying) the proposed method to DFA variants that have been proposed since the original DFA paper came out.